# JOOCI: A Framework for learning Comprehensive Speech Representations

## Abstract

Information in speech can be divided into two categories: "what is being said" (content) and "how it is expressed" (other). Current state-of-the-art (SOTA) techniques model speech at fixed segments, usually 10-25 ms, using a single embedding. Given the orthogonal nature of other and content information, attempting to optimize both within a single embedding results in suboptimal solutions. This approach divides the model's capacity, limiting its ability to build complex hierarchical features effectively. In this work, we present an end-to-end speech representation learning framework designed to jointly optimize the "other" and "content" information (JOOCI) in speech. By using separate learnable parameters, JOOCI addresses this optimization challenge by modeling other and content information independently. Our results show that JOOCI consistently outperforms other SOTA models of similar size (100 million parameters) and pre-training data used (960 hours) by a significant margin when evaluated on a range of speech downstream tasks in the SUPERB benchmark, as shown in Table 1. Code and models are available at TBA.

## 1 Introduction

Self-supervised learning (SSL) has played a significant role in learning high-level representations of text (Brown et al., 2020), vision (Alexey, 2020), and audio (Baevski et al., 2020; Mohamed et al., 2022; Défossez et al., 2022) data. In this work, we focus on learning high-level representations from raw speech. These learned representations are used as input features for various downstream tasks that require effective modeling of the "content" (what is being said) and "other" information (closely related to how it is expressed) present in speech (Yang et al., 2024; Borsos et al., 2023; Wang et al., 2023). The SUPERB (Yang et al., 2021) benchmark is designed to evaluate the generalization capabilities of different methods by evaluating the learned representations on a range of downstream tasks such as automatic speech recognition (ASR), phoneme recognition (PR), speaker identification (SID), emotion recognition (ER), and voice conversion (VC). ASR and PR performance depends on effective modeling of content information, while SID and ER rely on effective modeling of other information. The VC task requires the model to not only learn both information but also disentangle them.

Current state-of-the-art SSL methods, such as WavLM (Chen et al., 2022), on the SUPERB benchmark share two key similarities. First, these methods first downsample and embed raw speech into fixed segments, typically 25 ms, using a CNN encoder followed by a powerful transformer encoder [1] to learn contextual embeddings for each segment. Second, masked prediction loss (MPL) (Hsu et al., 2021; Chen et al., 2022) is used during pre-training.

However, two key limitations exist in current SOTA methods:

- Firstly, using a single embedding to learn other and content information results in suboptimal representations for both, due to the orthogonal nature of content and other information, which makes it difficult for the optimization algorithm to jointly learn both within a single embedding [2].

---

[1] Transformer encoder and model are used interchangeably.

[2] Perfect features for an ASR system should only encode content information i.e., what is being said and forget about everything other i.e., how it is expressed.

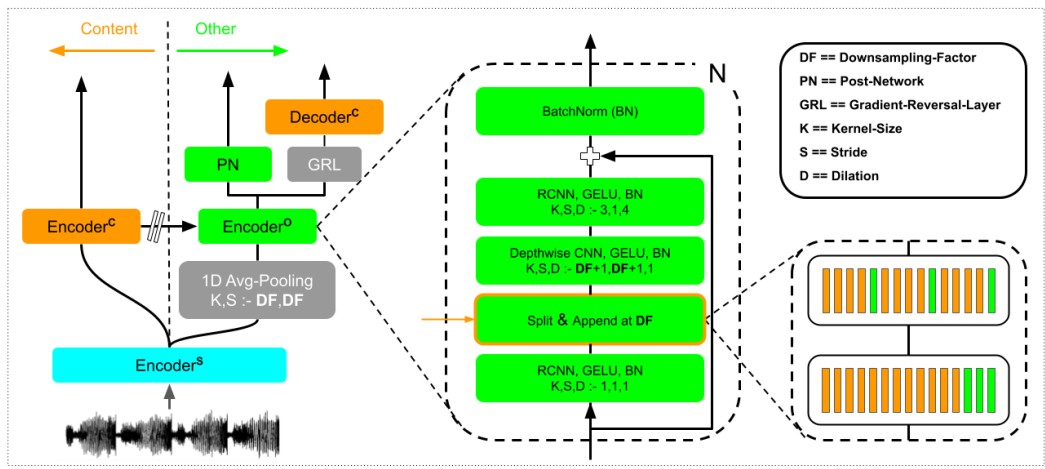

Figure 1: Our proposed JOOCI Method. A raw audio is passed through a shared encoder. The output is passed to the two separate encoders to model the other and content information. Other encoder can attend to the content encoder embeddings but the gradients do not flow from other encoder to content encoder. Better viewed in color.

- The second limitation is the MPL loss. (Yadav et al., 2023) showed that it maximizes content information learned during pre-training, which inadvertently minimizes the other information, particularly speaker-related information. This indicates that MPL alone is not enough for learning other information. Additionally, (Feng et al., 2022) demonstrated that models pre-trained with MPL encode speaker (other) information in the silent parts of audio, further corroborating this issue.

For example, WavLM uses data augmentation for learning other (speaker) information. The authors observed that the model divides its total modelling capacity/layers in two, with later layers learning content and earlier layers learning other information. This ultimately prevents the model from fully leveraging all layers to build the complex, hierarchical representations characteristic of deep learning.

Furthermore, when a method maximizes content information to the extreme to enhance the performance of ASR and PR tasks, such as MS-HuBERT (Yadav et al., 2024), the strategy of dividing layers to encode content and other information, as used in WavLM, becomes ineffective. Therefore, there is a need for a better framework capable of jointly learning other and content information in speech by utilizing all layers to build complex, hierarchical representations. In this paper, we present a framework to jointly optimize the "other" and "content" information (JOOCI). JOOCI models other and content information using separate learnable parameters and distinct loss functions, differentiating it from the previous SOTA method, WavLM (Chen et al., 2022), for learning comprehensive speech representations.

Our contributions can be summarized as follows:

1. We present JOOCI, a framework designed to jointly learn other and content information using separate learnable parameters and distinct loss functions. We achieve SOTA performance on the SUPERB benchmark, as shown in Table 1.

2. Adapters are one way to adapt the behavior of the pre-trained model (Chen et al., 2023b;c) for new tasks. In Section 3.2, we discuss in detail how JOOCI differentiates itself from adapters and why it is a more efficient choice.

## 2 JOOCI

In this section, we explain the proposed JOOCI framework in detail, covering its various components and the training criteria used. The complete JOOCI framework is illustrated in Figure 1.

## 2.1 COMPONENTS

**Shared Encoder ($\mathbf{E}^S$) :** The shared encoder serves a dual purpose i.e., it downsamples the raw speech and provides an embedding layer for both the content and other encoders. Inspired by previous work (Baevski et al., 2020; Hsu et al., 2021), we use seven blocks of temporal convolution layers which results in a down-sampling factor of 320x [3]. For more details of shared encoder please refer to the original work (Hsu et al., 2021; Yadav et al., 2024) that we have followed.

**Content Encoder ($\mathbf{E}^C$) :** The content encoder models "what is being said", i.e., the content present in speech. Following prior studies (Hsu et al., 2021; Yadav et al., 2024), the output from the shared encoder is randomly masked and passed through a series of $n$ self-attention based transformer encoder layers to model the content information. For more details related to the content encoder please refer to the original work (Yadav et al., 2024) that we follow.

**Other Encoder ($\mathbf{E}^O$) :** The other encoder models non-linguistic information present in speech. First, the shared encoder output is further downsampled using 1D average pooling layer with a kernel size and stride set to DF, where DF is the downsampling factor. The updated downsampled output is then passed through $n$ other encoder blocks, each of which consists of the following transformations:

- 1D Res2Net block (Gao et al., 2019) with a kernel size, stride and dilation all equal to 1. To increase the overall non-linearity of the module.
- A split and append layer.
- A depthwise 1D CNN layer [4] with a kernel size, stride and dilation equal to DF+1, DF+1, and 1 respectively.
- Another 1D Res2Net block with a kernel size, stride and dilation equal to 3, 1, and 4 respectively.
- A residual connection, followed by a 1D batch normalization (BN) layer

The split and append layer enables information from the content encoder to flow into the other encoder during the forward pass. No gradients flow from other encoder to content encoder during the backward pass. Therefore the other encoder can extract useful information, if any, from the powerful transformer content encoder. Specifically, content encoder embeddings are split into $m$ groups of size DF, and one embedding from the other encoder is appended at the end of each group sequentially. Finally, all groups are concatenated, increasing the total length, as shown in Figure 1. To reduce the input back to its original length, a depthwise CNN layer is applied with a kernel size and stride equal to DF+1.

**Post Network (PN) :** The post network performs three main functions crucial for training the other encoder using the student-teacher framework. 512D embeddings are extracted from the RDINO (Chen et al., 2023a) model acting as a teacher, for each utterance.

- An attentive statistical pooling (ASP) layer, similar to (Okabe et al., 2018), which aggregates variable-length inputs.
- A 1D BN layer.
- A fully connected (FC) layer with an output dimension of 512. Used only during the pre-training stage.

**Content Decoder :** Similar to (Ao et al., 2022), we train a single layer decoder using pseudo targets as ground truth. A gradient reversal layer (GRL) is applied before passing the gradients to the other encoder during back-propagation. The role of content decoder is to discourage the other encoder from learning features necessary for solving tasks that require content information. The content decoder and the GRL is used during the pre-training stage only.

## 2.2 TRAINING

During pre-training, an audio utterance $\boldsymbol{X} = x_1, x_2, \ldots, x_t$ is transformed using the shared encoder to produce embeddings ($\boldsymbol{E}$). These embeddings serve as inputs to both the content and other en-

---

[3]Each embedding represents 20ms of audio

[4]The number of groups equal to the other embedding dimension.

coders, which transforms them even further separately. (i) Before passing the embeddings to the content encoder, approximately 50% of the embeddings are masked and replaced with the masked embedding $[M]$. (ii) And before passing the embeddings to the other encoder, they are transformed using a 1D average pooling layer with kernel size and stride equal to the DF, reducing the embeddings again by a factor of DF [5].

**Content Loss (CL):** The Multicluster Masked Prediction Loss (MMPL), inspired by (Yadav et al., 2024), calculates the masked prediction loss (MPL) across multiple layers of the content encoder using six sets of different pseudo labels. These layers are selected at regular intervals between the last and an intermediate layer. MMPL is the sum of MPL over a set [6] = ( layer number, number of pseudo labels ) as shown in Equation 1.

$$L_{CL} = \sum_d (MPL) \tag{1}$$

where $d$ is a dictionary indicating which pseudo label set corresponds to which transformer layer. And MPL is computed only at the masked indices, as shown in Equation 2.

$$L_{MPL} = \frac{1}{\mathcal{M}} \sum_{i=1}^{\mathcal{M}} \frac{exp\left(\text{sim}(Ah_i, \mathbf{e}_c)/\tau\right)}{\sum_{c'=0}^{C-1} exp\left(\text{sim}(Ah_i, \mathbf{e}_{c'})/\tau\right)} \tag{2}$$

Here, $\mathcal{M}$ is the masked indices, $A$ is the projection matrix, $h_i$ is the content encoder embedding, $e_c$ and $e_{c'}$ represent the correct and incorrect embeddings for the pseudo labels, $sim(,)$ computes cosine similarity between two vectors, and $\tau$ a scaling factor for the logits. For further details, refer to (Yadav et al., 2024; Hsu et al., 2021).

**Other Loss (OL) :** The other encoder is trained using a student-teacher framework, where RDINO (Chen et al., 2023a), a self-supervised model trained to learn speaker-discriminative embeddings, serves as the teacher. RDINO has 22.74 million parameters, significantly more than the 3.52 million parameters in the other encoder. We maximize the cosine similarity between the output of the PN (Student) module and RDINO (Teacher) embeddings for a given utterance. Similar to the BYOL approach (Grill et al., 2020) we only use the positive pairs for loss calculation. The full objective is shown in Equation 3.

$$\mathcal{L}_{\text{OL}} = 1 - \text{sim}(Student^{PN}, Teacher^{RDINO}) \tag{3}$$

**Regularization Loss (RL) :** Following (Ao et al., 2022), we train a transformer decoder to predict pseudo labels, using cross-entropy loss as defined in Equation 4. The GRL scale the gradients during backpropagation by a factor of 1/10, preventing interference with the other loss.

$$\mathcal{L}_{\text{(RL)}} = -\sum_{t=1}^{T} \sum_{i=1}^{V} y_{t,i} \log \hat{y}_{t,i} \tag{4}$$

Where $T$ is the total number of time steps , $V$ is the size of the vocabulary, $y_{t,i}$ and $\hat{y}_{t,i}$ are the true one-hot encoded pseudo labels and predicted probability for the $t^{\text{th}}$ time step and $i^{\text{th}}$ word in the vocabulary respectively.

**Total Loss :** The overall training objective is the sum of all previous losses, as defined in Equation 5. Since JOOCI uses separate learnable parameters, the losses are summed directly without requiring additional hyperparameter tuning.

$$L_{Total} = L_{CL} + L_{OL} + L_{RL} \tag{5}$$

---

[5]Each embedding now represents DF times 20ms of audio for e.g., if DF is set to 10 then the other encoder is processing 200ms of audio.

[6]The set is of size 6, similar to the MS-HuBERT (Yadav et al., 2024).

| Method | #Params | Corpus | Speaker | | Content | | Semantics | | | ParaL |
|---|---|---|---|---|---|---|---|---|---|---|
| | | | SID | ASV | PR | ASR | IC | SF | | ER |
| | | | Acc ↑ | EER ↓ | PER ↓ | WER ↓ | Acc ↑ | F1 ↑ | CER ↓ | Acc ↑ |
| FBANK | 0 | - | 8.5e-4 | 82.01 | 9.56 | 23.18 | 9.10 | 69.64 | 52.94 | 35.39 |
| modified CPC | 1.84M | LL 60k hr | 39.63 | 12.86 | 42.54 | 20.18 | 64.09 | 71.19 | 49.91 | 60.96 |
| HuBERT | 94.68M | LS 960 hr | 81.42 | 5.11 | 5.41 | 6.42 | 98.34 | 88.53 | 25.20 | 64.92 |
| WavLM | 94.70M | LS 960 hr | 84.51 | 4.69 | 4.84 | 6.21 | **98.63** | **89.38** | **22.86** | **65.94** |
| JOOCI (Ours) | 109M | LS 960 hr | **90.79** | **4.15** | **4.25** | **5.35** | 98.42 | 88.77 | 23.74 | 65.24 |
| WavLM + | 94.70M | **Mix 94k hr** | 89.42 | 4.07 | 3.92 | 5.59 | 99.0 | 90.58 | 21.20 | 68.65 |

Table 1: UNIVERSAL SPEECH REPRESENTATION EVALUATION ON SUPERB BENCH-MARK. The performance on the content and semantic tasks can be further increased as shown in Table 4. To provide readers with a broader understanding of how performance scales with data size, we also include results for WavLM+, which has been trained on 100 times more data. **However, WavLM+ should not be directly compared to JOOCI, as the pre-training data disparity makes such comparisons inappropriate.**

# 3 EXPERIMENTS

## 3.1 EXPERIMENTAL SETUP

**Pre-training Dataset :** We use the LibriSpeech 960-hour dataset (Panayotov et al., 2015), excluding transcriptions, for pre-training JOOCI consistent with other studies focusing on speech representation learning (Schneider et al., 2019; Chung et al., 2021; Hsu et al., 2021).

**Data Augmentation**: Given that (i) LibriSpeech is a read speech dataset recorded in clean environment free from real world noises and (ii) the amount of dataset is under 1000 hours. We augment the data very lightly, so not to interfere with the content encoder a lot, which would result in the model dividing its capacity simliar to WavLM (Chen et al., 2022). We augment only 12.5% of the audio samples using the MUSAN corpus (Snyder et al., 2015) with a high signal-to-noise ratio (SNR) in the ranges of [5, 15] for noise, [13, 20] for speech, and [5, 15] for music. For comparison, WavLM augments 50% of the audio with a SNR value in between [-5, 5]. Lastly, we also add Room Impulse Response (RIR) for reverberation to the selected audios similar to RDINO (Chen et al., 2023a).

**Pre-training :** We use 12 layers for the other and content encoder. For the decoder, we use one transformer decoder layer with 8 heads and 768 dimension. It is trained with a vocabulary size of 1005. Other encoder is trained using the student-teacher framework. Content encoder is trained exactly to MS-HuBERT (Yadav et al., 2024). A detailed description is provided in the Appendix.

**Shared Encoder and Content Encoder Initialization :** The shared encoder and content encoder of JOOCI are initialized with the pre-trained MS-HuBERT model weights (Yadav et al., 2024). This choice is based on MS-HuBERT's superior performance in modeling content information using the Multicluster Masked Prediction Loss (MMPL), which has been shown to achieve SOTA performance on content-based tasks.

**SUPERB :** To comprehensively analyze both the content and other types of information learned by JOOCI, we evaluated several tasks from the SUPERB (Yang et al., 2021) benchmark and compared our method with other SOTA methods of similar model size. The SUPERB benchmark is designed to compare models on their ability to learn comprehensive audio characteristics across a variety of downstream tasks including speaker recognition, content analysis, semantics, para-linguistics, and generation. For more details, see Yang et al. (2021). In short, SUPERB freezes the encoder and learns a weighted sum of all the layers to produce features for different downstream tasks.

**Fine-tuning :** No fine-tuning was performed. After pre-training, we discarded the content decoder and the fully connected (FC) layer from the post-network (PN) module and froze the parameters of JOOCI for all evaluation purposes in this work.

### 3.2 MAIN RESULTS

**Evaluating JOOCI on the SUPERB Benchmark :** Table 1 summarizes the performance of JOOCI on a range of downstream tasks from SUPERB benchmark. The results clearly indicate that JOOCI outperforms the current state-of-the-art (SOTA) models on the majority of tasks, except few where the margin is very less [7]. In particular, JOOCI maximizes the performance on speaker-related tasks such as speaker identification (SID) and speaker verification (ASV), and exhibits the lowest error rates in content-related tasks like phoneme recognition (PR) and automatic speech recognition (ASR). Overall, JOOCI's performance on the SUPERB benchmark reinforces our initial claim of it being a comprehensive framework for learning speech representations.

**Comparison with Adapters :** Instead of fine-tuning pre-trained models for every new downstream task, adapters (Chen et al., 2023b;c) offer a lightweight solution to adapt these models for new tasks. Adapters introduce a small number of additional parameters tailored for each specific task. Table 2 compares JOOCI with HuBERT fine-tuned using various adapters. A key limitation of adapters, however, is that for every new task, it requires a new forward pass through the pre-trained model. JOOCI stands out by overcoming this limitation, as both the other and content embeddings are extracted in parallel in a single forward pass. With (i) comparable performance to adapters and (ii) the advantage of requiring only one forward pass, JOOCI stands as a more efficient framework for learning comprehensive speech representations. Lastly, Adapters can be applied to JOOCI also.

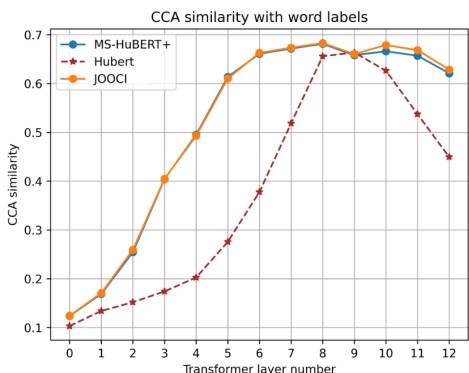

| Method | Params | SID ↑ | ASV ↓ | ASR ↓ | PR ↓ |
|---|---|---|---|---|---|
| HuBERT | | | | | |
| FT | 94.68M | 64.56 | 5.15 | 6.35 | 2.45 |
| Weighted-sum | 13 | 81.42 | 5.11 | 6.42 | 5.41 |
| HuBERT + Adapter (Chen et al., 2023c) | | | | | |
| Houlsby | 0.60M | 87.71 | 5.29 | 5.88 | 3.00 |
| CHAPTER | 4.67M | 91.56 | 4.95 | 6.22 | 2.95 |
| JOOCI (Ours) | | | | | |
| Weighted-sum | 13 | 90.79 | 4.15 | 5.35 | 4.25 |

Table 2: Comparison of JOOCI with HuBERT using adapters. FT stands for finetuning. Here 13 is not a mistake and the weighted-sum only uses 13 parameters for each of the 13 layers.

Figure 2: Studying the effect of data augmentation on the content encoder using CCA word label similarity. Higher the CCA similarity for more number of layers better the method is. Better viewed in color.

**Effect of Data Augmentation on the Content Encoder :** We studied the extent to which data augmentation alters the learned representations in the content encoder. Following the approach of (Pasad et al., 2021; 2023), we plotted Canonical Correlation Analysis (CCA) versus word label similarity, as shown in Figure 2. Interestingly, the similarity in the later layers increased, suggesting that even a small amount of data augmentation aids in learning more robust features. We hypothesize that this is due to the small size of the pre-training dataset. Based on these insights and the findings from Section 4.3, we selected layers 6 to 11 (six layers) that exhibited very high CCA word label similarity and re-ran the experiments on ASR, PR, SF, and IC tasks. This selection resulted in a performance boost on the content tasks, either outperforming or matching WavLM on semantic tasks, as shown in Table 4. Therefore, taking a weighted sum of layers that encode similar information is a more robust choice than using all layers.

| Layers | Acc ↑ | Importance of other encoder layers |
|--------|-------|-------------------------------------|
| | Weighted Sum | |
| 13 | 88.84 (89.20*) | |
| 6 | 89.10 (89.32*) |  |
| 3 | 88.87 (90.07) | |
| | Single Layer | |
| LastLayer | 90.24 | |
| + ASP | 89.49 | |
| + BN | 90.79 | Not Applicable |
| + FC | 88.80 | |

Table 3: Ablation results. Performance of the SID task for different configurations. Better viewed in color.

| Method | PR | ASR | SF | | IC |
|--------|-----|------|------|------|------|
| | PER | WER | F1 | CER | Acc |
| WavLM | 4.84 | 6.21 | 89.38 | **22.86** | **98.63** |
| JOOCI | 4.25 | 5.35 | 88.77 | 23.74 | 98.42 |
| JOOCI (6 - 11) | **4.19** | **5.20** | **89.61** | 23.32 | 98.61 |

Table 4: Ablation results. Performance of JOOCI when only using layers with high CCA similarity score.

# 4 ABLATION STUDY

## 4.1 INFORMATION ENCODED AT DIFFERENT LAYERS IN THE OTHER ENCODER FOR THE SID TASK

The expectation is that JOOCI's other encoder **SHOULD** build complex hierarchical features, with later layers being more important for solving tasks such as speaker recognition [8]. Table 3 shows the accuracy for the SID task across different configurations corroborating our claim for JOOCI leveraging all the layers to build hierarchical features for other information also. Surprisingly, When using a weighted sum of the other encoder's all the layers, the layers towards end (6-9) are assigned higher weight and not the last layer. To investigate this further, we use only the last six layers, without any performance drop, yet the same trend persisted. Next, we tried using only the last three layers, without any performance decline, which again showed the same pattern. Finally, using only the last layer results gives the best result. We hypothesize that this could be because of the softmax property used in the SUPERB benchmark for assigning weights to all the layers, which gives a uni-modal distribution. Using a setup similar to multi-head attention might give a clearer picture. We leave this for future work.

Next, applying an ASP layer followed by Batch Normalization (BN) layer to the output of the last layer boosts the performance even further. On the other hand, using the last fully connected (FC) layer resulted in a performance drop of approximately 2%. This performance decline when using the last layer, closest to the loss function, has been observed in vision model trained using SSL, where it is often recommended to drop the last layer after pre-training (Chen et al., 2020).

Based on these findings, we conclude that the model effectively constructs complex, high-level features by utilizing all layers for tasks requiring other (speaker) information. And in turn, JOOCI is able to jointly learn other and content information utilizing all the layers.

---

[7]WavLM used very big batch sizes and this could be the reason for the little margin gains compared to HuBERT and JOOCI.

[8]It has been shown in the literature that models using MPL encodes content information in the later layers and other (speaker) information in the earlier layers.

| Method | Corpus | SID Acc ↑ | PR PER ↓ | ASR WER ↓ | ER Acc ↑ |
|---|---|---|---|---|---|
| Teacher model (22.74 million) | | | | | |
| RDINO | VC2 2,442hr | 96.68 | - | - | 53.21 |
| Keeping Shared and Content Encoder Frozen + **NO** Data Augmentation | | | | | |
| JOOCI-C | LS 960hr | 75.40 | 4.17 | 5.32 | 62.05 |
| JOOCI-O | LS 960hr | 88.94 | - | - | 61.86 |
| JOOCI-O-DGRL | LS 960hr | 88.50 | - | - | 62.72 |
| Finetuning complete JOOCI setup + Data Augmentation | | | | | |
| JOOCI-C | LS 960hr | - | 4.25 | 5.35 | 65.27 |
| JOOCI-O-DGRL | LS 960hr | 90.79 | 99.02 | - | 64.38 |

Table 5: Ablation results. Data augmentation helps on the ER task as a lot. Surprisingly, for the ER task content encoder embeddings with data augmentation performs better than the other encoder embeddings. JOOCI-C means using the content encoder embeddings, JOOCI-O means using other encoder embeddings and JOOCI-O-DGRL means using other embeddings trained with the decoder and GRL layer.

## 4.2 FROZEN SHARED AND CONTENT ENCODER VS FINE-TUNED SHARED AND CONTENT ENCODER WITH DATA AUGMENTATION.

JOOCI consists of three major components: (i) content encoder (C), (ii) other encoder (O), and (iii) content decoder with GRL (DGRL). We explore different combinations of these components across four downstream tasks, which tests effective modeling of both content and other information. Table 5 presents our ablation results. First, we pre-train JOOCI by training only the other encoder (O) and keeping the shared and content encoder frozen, either with or without the DGRL module. Incorporating the DGRL module significantly boosts ER performance, although it slightly degrades SID performance. Overall, the inclusion of DGRL increases robustness, and thus we employ the DGRL in all subsequent JOOCI experiments.

Next, we study the effect of data augmentation during pre-training, when all parameters of JOOCI are trained. Data augmentation results in substantial performance gains on the SID and ER tasks. These improvements are likely due to the characteristics of the LibriSpeech pre-training dataset, consisting of read speech with minimal background noise. For the PR and ASR task, data augmentation leads to very minimal degradation, which is negligible when considering the significant gains on other tasks. Our hypothesis is that with larger and more diverse datasets, the dependence on data augmentation will diminish.

## 4.3 DISENTANGLEMENT PROPERTY OF JOOCI

Disentanglement is the ability of the model to represent multiple factors of variation in data as separate and independent latent variables or embeddings. These embeddings encode different aspects of the data without influencing each other.

Solving the voice conversion from any to any (VC-a2a) speaker requires effective disentanglement of the current speaker from the content information in speech. Our findings indicate that either using one layer or the layers with high CCA similarity results in improved performance compared to using all the layers, as demonstrated in Table 6. This shows that JOOCI is able to disentangle content from the other information in the layers with high CCA scores. Furthermore, Table 5 shows that the other encoder also is not encoding content information as the performance on the PR task is 100%. Both these observations prove that JOOCI is able to disentangle content and other information successfully.

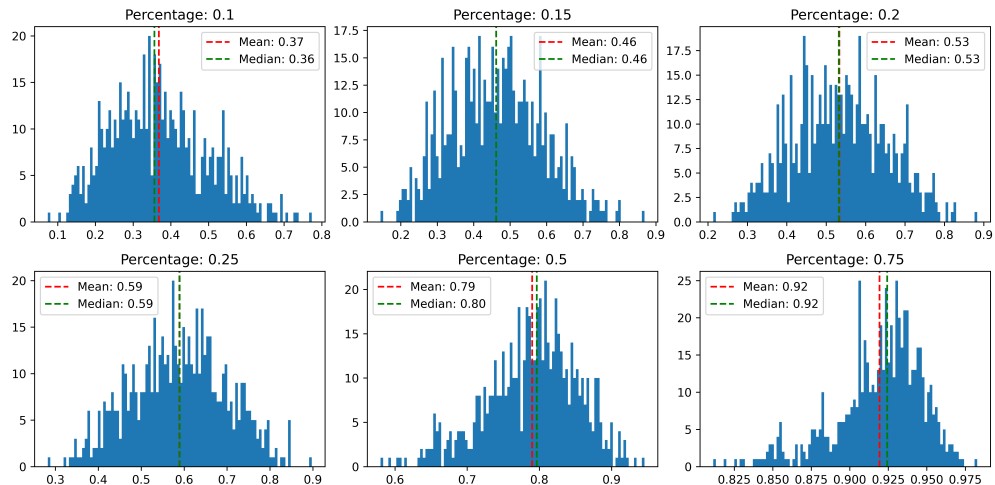

Figure 3: Ablation results. ASP layers uses attention weights, between 0 and 1, to create one embedding for a variable sequence length. Sorting the weights, we see that majority of weights were given to few indices. Each histogram plots the attention weight values for different percentage of indices used. As we can see 50% of the total attention weight is given to only the top 20% indices. Better viewed in color.

| Method | Speaker-Embedding | VC-a2a | | |
|---|---|---|---|---|
| | | MCD ↓ | WER ↓ | ASV ↑ |
| All the layers | | | | |
| HuBERT | d-vector | 9.19 | 3.4 | 23.25 |
| JOOCI | | 9.34 | 3 | 22 |
| one layer | | | | |
| HuBERT | d-vector | 8.49 | 3.3 | 66 |
| JOOCI | | 8.31 | 2.9 | 86.5 |
| Using later layers | | | | |
| HuBERT (8 - 9) | d-vector | 8.52 | 3.5 | 65.5 |
| JOOCI (6 - 11) | | 8.40 | 3.1 | 91.5 |
| JOOCI (6 - 11) | JOOCI | 8.12 | 3.1 | 83.00 |

Table 6: Ablation results. Using layers with high CCA scores (layer 6 to 11) is able to disentangle content information from the other information. We also use JOOCI other ASP output as speaker embeddings and find lower MCD, which shows more natural audio but lower speaker metric. In contrast, d-vector is a supervised trained system and (Wan et al., 2018) has 1.48 million parameters. Using JOOCI's other encoder as speaker embedding results in better MCD. But lower ASV could be because of small pre-training data used.

## 5 CONCLUSION

This paper presents JOOCI, a novel framework for learning comprehensive speech representations by jointly optimizing other and content information. JOOCI addresses the limitations of existing methods that struggle to encode both types of information effectively. By employing separate encoders and distinct training criteria for other and content information, JOOCI achieves state-of-the-art performance on the SUPERB benchmark, outperforming existing models across a range of downstream speech tasks, including speaker identification, speaker verification, phoneme recognition, and automatic speech recognition. The efficiency of JOOCI is also emphasized, as it extracts both other and content embeddings in a single forward pass, in contrast to adapter-based methods that require multiple forward passes. The ablation study highlights the significance of the DGRL module in enhancing the robustness and performance of JOOCI, particularly for the emotion recognition task.

| Encoder | SD (DER ↓) |
|---------|------------|
| label frame shift = 320 | |
| HuBERT | 5.88 |
| WavLM | 4.55 |
| JOOCI-C | 5.97 |
| JOOCI-O | 6.26 |

| Encoder | SD (DER ↓) |
|---------|------------|
| label frame shift = 3200 | |
| JOOCI-C | 7.18 |
| JOOCI-O | 7.23 |

Table 7: Limitations. Performance on the SD task. We can not compare it with HuBERT or WavLM because other encoder's resolution is 10 times of them. We copy the other encoder 10 times for each index to decrease the label frame shift (320). And we increase the content encoder's label frame shift (3200) by a factor of 10 using mean pooling for comparison.

Additionally, data augmentation during pre-training proves beneficial for speaker identification and emotion recognition tasks, likely due to the size of the LibriSpeech dataset.

## 6 LIMITATIONS AND FUTURE WORK

**Speaker Diarization (SD) Task :** The SD task involves determining who spoke when in an audio recording. It requires the system to segment the audio and assign speaker labels to each segment, effectively separating multiple speakers within a conversation or audio stream. Unexpectedly, despite the other encoder's strong performance on tasks like SID and ASV, it falters in the SD task compared to the content encoder. The results are shown in Table 7. The degradation maybe because of using the one embedding, during pre-training other encoder, for the entire utterance. On the other hand, data augmentation proves to be highly effective for learning fine-grained, discriminative features as shown by better performance from the content encoder.

**Better Pre-training Methodology :** (i) Employing a different pre-training methodology to learn fine-grained, discriminative features for the other encoder, such as using neural codec latent variables as training targets (Défossez et al., 2022). (ii) To improve content modeling, utilizing the content decoder to also train the content encoder.

**Multi-Talker ASR :** JOOCI is well-suited for multi-talker ASR without requiring any external modules. It can be easily adapted by (i) using other embeddings to classify similar speakers using unsupervised clustering, and (ii) applying the resulting classification as a mask to segment content embeddings, which can then be used for the ASR task.

### ACKNOWLEDGMENTS

Hemant Yadav is supported by Microsoft Research India PhD Fellowship program. Rajiv Ratn Shah is partly supported by the Infosys Center for AI, the Center of Design and New Media, and the Center of Excellence in Healthcare at IIIT Delhi.

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

# A    APPENDIX

## A.1    PRE-TRAINING

**Parameter count comparison :** Table 8 shows the parameter count of different components of JOOCI during pre-training and inference.

**Pre-training :** Pre-training involves either a frozen content encoder or an unfrozen content encoder.

In the frozen content encoder case, JOOCI is trained for 50,000 iterations on 4 GPUs, with each GPU processing up to 375 seconds of audio, resulting in a total of 1500 seconds of audio per iteration. The learning rate is set to 5e-4, using the first 5,000 steps for warm-up. All other settings are consistent with those used in MS-HuBERT Yadav et al. (2024).

In the second case, pre-training continues from the first case [9] for an additional 100,000 iterations, using a learning rate of 5e-5 across 6 GPUs, with each GPU handling up to 200 seconds of audio, yielding a total of 1200 seconds of audio per iteration. The first 1,000 steps are used for warm-up updates. All other settings are consistent with those used in MS-HuBERT Yadav et al. (2024).

|  | Training | Inference |
|---|---|---|
| HuBERT | 94.70M | 94.70M |
| Shared encoder & Content encoder | 96.18M | 94.70M |
| Other encoder & PN | 3.52M | 3.12M |
| Content decoder | 9.3M | - |
| JOOCI (Ours) | 109M | 97.82 M |

Table 8: Parameter count for different components of the JOOCI framework compared to HuBERT millions.

## A.2    SUPERB BATCH SIZE COMPARISON

Table 9 shows the comparison of batch sizes used by WavLM and JOOCI. And what is used in the SUPERB benchmark.

| Task | WavLM | JOOCI | SUPERB |
|---|---|---|---|
| SID | 512 | 32 | 32 |
| ASR | 128 | 32 | 32 |
| IC | 128 | 32 | 32 |
| SF | 128 | 32 | 32 |
| ER | 32 | 32 | 32 |

Table 9: Batchsize used by WavLM and JOOCI and the SUPERB benchmark default.

---

[9]Using the data augmentation.

