# OpenReview forum: "JOOCI: A FRAMEWORK FOR LEARNING COMPREHENSIVE SPEECH REPRESENTATIONS"
_ICLR.cc/2025/Conference — Submitted to ICLR 2025_

### Official Review · Reviewer_rAqx · 2024-11-03

**Soundness:** 3
**Presentation:** 3
**Contribution:** 2
**Rating:** 5
**Confidence:** 4

**Summary:**

The paper proposes to disentangle the content "what is being said" and other “how it is expressed” information present in the speech data. The paper proposes the JOCCI framework, which uses two submodules, focused on maximizing the content and the other information. The content module is trained with a self-supervised objective whereas, the other module is optimized with a teacher-student objective. A regularization loss is added to minimize the information overlap in the two submodules.

**Strengths:**

1) The paper is well-written and addresses an important challenge of disentangling content and other information in the speech representation.

2) The model performs well on the SUPERB benchmark and outperforms HuBERT and wavLM.

**Weaknesses:**

1) The baseline comparisons are limited. There have been other attempts to remove other information such as speaker information from the self-supervised representations such as contentvec[1] and SPIN[2]. Even the MS-HuBERT model used for initializing JOCCI is missing from Table 1.

2) JOCCI relies on a pretrained method RDINO for training the other encoder whereas baseline methods such as HuBERT do not.

[1] “Contentvec: An improved self-supervised speech representation by disentangling speakers

[2] Self-supervised Fine-tuning for Improved Content Representations by Speaker-invariant Clustering

**Questions:**

1) How does the JOCCI compare to the contentvec, SPIN, and Data2vec models?

2) What is the impact of initialization on the model performance e.g. random vs Ms-HuBERT initialization

3) Can the model be trained with just the GRL loss and without the RDINO teacher?

---

> ### Author Response · Authors · 2024-11-14
> **Rebuttal on JOOCI’s Framework and Comparisons to Existing Models. JOOCI is not doing any disentanglement.**
>
> ### We thank the reviewer for their thoughtful comments and the opportunity to address their questions and observations.
>
> * Firstly, we would like to clarify that JOOCI does not aim to remove other information from the content or employ a disentanglement module. Instead, JOOCI optimizes both content and other information jointly, using separate paths/encoders to maximize the representation quality without attempting a strict separation observed in WavLM.  In fact, the concept of disentanglement is only mentioned in Section 4.3 of the paper (in the second-to-last section before the conclusion) as part of an ablation study on understanding learned representations.
>
> * We are seeking further clarification regarding the statement: "JOCCI relies on a pretrained method RDINO for training the other encoder, whereas baseline methods such as HuBERT do not." Many state-of-the-art methods, such as Vall-E[1], also use pretrained models (Encodec) as teachers. We would appreciate the reviewer’s insight on how this is considered a weakness in the context of our work.
>
> * Regarding the comparison with ContentVec, SPIN, and Data2Vec, we believe these methods serve different purposes from JOOCI, focusing primarily on content representation without optimizing for both content and other information. Our content encoder is initialized from MS-HuBERT, which, as demonstrated on the SUPERB benchmark, outperforms ContentVec, SPIN, and Data2Vec in the phoneme recognition (PR) task. JOOCI, however, is designed to perform competitively on both PR and speaker identification (SID) tasks, demonstrating its ability to effectively handle both types of information. A direct comparison would be WavLM.
>
> * ContentVec and SPIN are models that fine-tune other pre-trained models, whereas JOOCI focuses solely on the pre-training stage. Given this distinction, we are unsure how including these models would contribute to strengthening the claims of our paper.
>
> * We would appreciate the reviewer’s perspective on how evaluating the impact of random versus MS-HuBERT initialization, as well as training the model with just the GRL loss and without the RDINO teacher, might help corroborate the effectiveness of JOOCI.
>
> We hope this addresses the reviewer's concerns, and we appreciate any further feedback that may help clarify or strengthen our presentation.
>
>
> [1] Neural Codec Language Models are Zero-Shot Text to Speech Synthesizers

---

> > ### Comment · Reviewer_rAqx · 2024-11-20
> >
> > 1) "Firstly, we would like to clarify that JOOCI does not aim to remove other information from the content..."
> >
> > Line 153 in the paper: "The role of content decoder is to discourage the other encoder from learning features necessary for solving tasks that require content information" so the GRL is used to minimize the content information in the other encoder explicitly. The paper content and your claim here about not removing information across the encoder seem contradictory to me.
> >
> > 2) The paper claims that JOOCI is more data efficient than WavLM and HuBERT. By using a pretrainined model RDINO, trained on the VOXceleb dataset JOCCI model effectively sees more data than HuBERT/WavLM. What if I train a model initialized with HuBERT and use WavLM-large as a teacher but only use Librispeech 960 hours to pretrain my model, it would be unfair to call it more data efficient than HuBERT trained from scratch.
> > Another reason is the simplicity of adapting the JOOCI to a new dataset/language, JOCCI requires MS-HuBERT to initialize and then RDINO  for the other encoder to be trained in the new language whereas HuBERT/WavLM requires just one model to be trained.
> >
> > JOOCI performs better than HuBERT/WvLM: yes, is it more data efficient: No
> >
> > 3) "Regarding the comparison with ContentVec, SPIN, and Data2Vec, we believe these methods serve different purposes from JOOCI, focusing primarily on content representation without optimizing for both content and other information. Our content encoder is initialized from MS-HuBERT, which, as demonstrated on the SUPERB benchmark, outperforms ContentVec, SPIN, and Data2Vec in the phoneme recognition (PR) task. JOOCI, however, is designed to perform competitively on both PR and speaker identification (SID) tasks, demonstrating its ability to effectively handle both types of information. A direct comparison would be WavLM."
> >
> > Since the MS-HuBERT model performance is not included, JOCCI may perform worse than MS-HuBERT on some of the PR tasks as it also focuses on the speaker tasks. Including MS-HuBERT/data2vec performance could help readers understand the trade-off between focusing on PR tasks vs both phone and speaker tasks. Why is MS-HuBERT not included?
> >
> > Again it does not matter if MS-HuBERT outperforms ContentVec, SPIN, and Data2Vec The paper focuses on JOCCI and not on MS-HuBERT so the comparisons should be with JOCCI and the paper does not provide results supporting that JOCCI outperforms MS-HuBERT.
> >
> > 4) "ContentVec and SPIN are models that fine-tune other pre-trained models, whereas JOOCI focuses solely on the pre-training stage. Given this distinction, we are unsure how including these models would contribute to strengthening the claims of our paper."
> >
> > ContentVec and SPIN initialize their models with HuBERT and then finetune them with their SSL objective similar to what is being done here: MS-HuBERT for initialization and then SSL objective that focuses on separating the content and other information. Content/SPIN also focuses on the pretraining stage. Could you please give more details on how these approaches differ and ContentVec and SPIN do not focus on the pretraining?
> >
> > 5) "We would appreciate the reviewer’s perspective on how evaluating the impact of random versus MS-HuBERT initialization, as well as training the model with just the GRL loss and without the RDINO teacher, might help corroborate the effectiveness of JOOCI."
> >
> > These changes show how stable the training objective is and how much each part contributes to the performance. For example, no GRL loss could be a good experiment to support your claim that JOCCI focuses on maximizing the information in each encoder and not on minimizing the information across encoders.

---

> > > ### Author Response · Authors · 2024-11-25
> > > **Response.**
> > >
> > > ### We thank the reviewer for their comments and the opportunity to address their questions and observations.
> > >
> > > *  Line 153 in the paper: The role of content decoder.......
> > >     * we would like to clarify that the GRL module serves primarily as a regularizer rather than a disentanglement module. Its purpose is to ensure that the other encoder does not overly specialize in speaker information alone. As demonstrated in Table 5, the GRL module results in a trade-off, slightly reducing SID performance while improving ER performance, highlighting its role as a balancing mechanism rather than a strict disentanglement tool.
> > >     *  Additionally, It discourages content information from other encoder and not the other way around as proposed in ContentVec and SPIN. We would like to emphasize that JOOCI does not remove any information from the content encoder; it simply optimizes it using the MMPL loss similar to MS-HuBERT or HuBERT.
> > >
> > > * The paper claims that JOOCI is more data efficient than WavLM and HuBERT......
> > >   * We kindly ask the reviewer to clarify where in the paper we claim that JOOCI is more data-efficient than HuBERT or WavLM, as we would like to better understand the source of this impression. On the contrary, We focus on the framework's ability to jointly optimize content and other information rather than to emphasize data efficiency as a key claim.
> > >   * Regarding the suggestion of using WavLM-large as a teacher and HuBERT-base as a student, we appreciate the perspective. However, such an approach would inherit the same limitations as the original HuBERT or WavLM frameworks, where the encoder’s capacity is split between the initial and later layers. In contrast, JOOCI adopts a fundamentally different framework by separately optimizing content and other representations. For instance, using the later layers of WavLM-large to train the content encoder and the initial layers to train the other encoder ensures that both types of information are optimized without dividing the encoder’s capacity within a single framework. This joint optimization is not possible by using a single encoder as in HuBERT or WavLM.
> > >
> > > * JOOCI performs better than HuBERT/WvLM: yes, is it more data efficient: No
> > >   * While it may be valid to argue that JOOCI benefits from leveraging RDINO as a pre-trained component. We request the reviewer to view JOOCI as a progression in the framework for learning comprehensive speech representations. Much like comparing GPT-3 to GPT-2 considers advancements in both scale and approach, JOOCI represents a next step in optimizing both content and other information jointly, which sets it apart from prior frameworks like HuBERT or WavLM.
> > >
> > > * Since the MS-HuBERT model performance is not inc......
> > >   * When the shared and content encoders are frozen, the results for the content based tasks are effectively as MS-HuBERT. For the reader’s reference, the corresponding performance on ASR and PR tasks is provided in Table 5. When using little data augmentation, it results in little improvements for other tasks and little degradation for content tasks. To ensure clarity, we will revise the table caption in the manuscript to explicitly highlight this equivalence.
> > >
> > > * Again it does not matter if MS-HuBERT ou....
> > >   * We would like to emphasize that the goal of JOOCI is not to maximize performance on content-based or speaker-based tasks individually but to optimize both jointly, similar to the aim of WavLM in learning comprehensive speech representations. However, while WavLM achieves this by dividing the encoder's capacity across layers, JOOCI takes a different approach by employing separate paths for these orthogonal types of information. The results presented in the paper demonstrate that JOOCI is a more effective choice for achieving this balance, offering improved representation quality for both content and non-content tasks. Table 5 shows the comparison as explained earlier.
> > >
> > > * ContentVec and SPIN initialize their.....
> > >   * To clarify, while both ContentVec and SPIN rely on HuBERT initialization, JOOCI does not require such initialization. Instead, using MS-HuBERT for initialization in JOOCI is a choice made primarily to save computational resources. To further illustrate this, ContentVec and SPIN could also use JOOCI's content encoder for initialization instead of HuBERT, which highlights that JOOCI operates at an earlier stage than ContentVec and SPIN in the pre-training process. Additionally, while ContentVec and SPIN explicitly aim to maximize performance on content-based tasks, JOOCI does not make this trade-off. JOOCI is designed to optimize both content and non-content information jointly, without having to prioritize one over the other.

---

> > > > ### Author Response · Authors · 2024-11-25
> > > > **Response (Contd).**
> > > >
> > > > * These changes show how stable t....
> > > >   * We would like to clarify that the impact of removing the GRL module on the other encoder is shown in Table 5. As demonstrated, the GRL module acts as a regularizer: its presence reduces performance on the SID task but improves performance on the ER task.  GRL module does not and can not affect the learning of content encoder because there is NO gradient flow, during backprop, from the other encoder to the content encoder. And thus other encoder can not emphasize  the content encoder and vice-versa.
> > > >
> > > >
> > > > We hope this explanation clarifies the role of the GRL module and its effects on model performance.

---

> > > > > ### Comment · Reviewer_rAqx · 2024-11-27
> > > > >
> > > > > Would it be correct to say that the GRL module tries to remove the content information from the other encoder?
> > > > > "Its purpose is to ensure that the other encoder does not overly specialize in speaker information alone" but the RDINO teacher is trained to generate speaker embeddings. How does using RDINO as a teacher for other encoders ensures that the other encoder does not overly specialize in speaker information alone?
> > > > >
> > > > > Line 22: "pre-training data used (960 hours) by a significant margin when evaluated on a range of speech downstream tasks in the SUPERB benchmark" in the abstract implies JOCCI is better than other models trained on the same amount of pretraining data. It is not true because JOCCI uses RDINO as a teacher which is trained on additional data (2.5K hours).
> > > > >
> > > > > "To clarify, while both ContentVec and SPIN rely on HuBERT initialization, JOOCI does not require such initialization. Instead, using MS-HuBERT for initialization in JOOCI is a choice made primarily to save computational resources." So both methods use pretrained models as initialization, what makes ContentVec and SPIN "rely" on HuBERT initialization and JOCCI doesn't rely on the initialization? Most models start from pretrained models nowadays for computational reasons.
> > > > >
> > > > > "To further illustrate this, ContentVec and SPIN could also use JOOCI's content encoder for initialization instead of HuBERT, which highlights that JOOCI operates at an earlier stage than ContentVec and SPIN in the pre-training process." JOCCI can also use HuBERT/SPIN/ContentVec as initialization. ContentVec, SPIN and JOOCI are all pertraining methods that start from an SSL model and further finetune it, what do you mean by " JOOCI operates at an earlier stage than ContentVec and SPIN in the pre-training process"? I would argue the opposite, JOCCI needs MS-HuBERT which needs HuBERT as initialization whereas ContentVec, and SPIN require one less stage of pretraining to get to the final model.

---

> ### Author Response · Authors · 2024-11-27
> **Response.**
>
> ### We thank the reviewer for their comments and the opportunity to address their questions and observations.
>
> * Would it be correc.....
>     * If it can be proved that the pseudo labels used accurately represent content information without including other information (e.g., speaker), then yes, the GRL module can be seen as attempting to remove only content information from the other encoder. The results shows otherwise, since using it also degrades the speaker information as shown in Table 5. We will update the writing to make this clearer for future readers.
>     * Regarding the use of the RDINO teacher, it does not explicitly ensure that the other encoder avoids specializing solely in speaker information. However, when the GRL module is applied, we observe that the other encoder balances its focus towards the ER task though at the cost of reduces performance on the speaker based tasks.
>
> * Line 22: "pre-training data u.....
>     * Thank you for bringing this important point to our attention. RDINO, is used as a teacher to extract speaker embeddings, which is trained on 2.5k data and is the reason for improved performance on the speaker based tasks such as SID and ASV as pointed out by the reviewers. And therefore, saying that comparing with WavLM (960 hours) might not be valid or true.
>     * To address this concern, we kindly request the reviewer to refer to Table 1, specifically the last row comparing WavLM+, which is trained on 94,000 hours of data (compared to just 960 + 2500 hours). The comparable performance on SID (a little improvement) and ASV (a little degradation) demonstrates that the gains are not solely due to data volume but also reflect JOOCI's ability to jointly optimize content and other information effectively. We will revise the abstract to clarify this distinction to prevent any potential misunderstanding.
>
> * To clarify, while ....
>     * Thank you for the clarification regarding ContentVec and SPIN. As pointed by the reviewers,  HuBERT/SPIN/ContentVec can be used as initialization for JOOCI's content encoder. But again, MS-HuBERT is a overall a better choice when compared to the above three, looking at their PR and ASR task performance.  The results for HuBERT/SPIN/ContentVec are shown in SPIN paper ( https://arxiv.org/pdf/2305.11072)  Table 1. MS-HuBERT's performance on PR and ASR task is 4.17 and 5.32 respectively as shown in Table 5.
>     * Furthermore, SPIN can be applied on MS-HuBERT to further improve its content representation since the  phonetic content resides in top layers which is not the case of data2vec as explained by the authors of SPIN paper in section 3.2. Therefore we request the reviewer to see MS-HuBERT similar to HuBERT than SPIN or ContentVec. That is why we made the observation that MS-HuBERT should be seen as an earlier stage of ContentVec or SPIN. On the other hand, when MS-HuBERT is used as initialization for JOOCI's content encoder, it is continued pre-trained using the similar MMPL loss. It can be simply frozen, for which the results are shown in Table 5 (Keeping Shared and Content Encoder Frozen + NO Data Augmentatio).
>
> * To further illustrate thi....
>     * We would like to address a misunderstanding: MS-HuBERT is not initialized with HuBERT instead HuBERT 1st iteration is only used to get cluster ids rather than starting from Fbanks to save computational resources.
>
> --------------------------------------------------------------------------------------------------------------------------------------
> #### Lastly, we respectfully maintain that ContentVec and SPIN have different goals compared to JOOCI. Both focus solely on maximizing content information, while JOOCI aims to jointly optimize both content and non-content representations, aligning more closely with WavLM's objectives.
> #### That said, we understand that it might be valuable to some reader in discussing the similarities and differences between these methods in detail. We will include a discussion section in the Appendix, in the revised version to elaborate on these points and provide the SUPERB benchmark numbers for ContentVec and SPIN for better clarity.

---

> ### Author Response · Authors · 2024-11-29
> **Updated response on the weaknesses.**
>
> * **Reviewer:** "The baseline comparisons are limited. There have been other attempts to remove other information such as speaker information from the self-supervised representations such as contentvec and SPIN."
>     * **Response:** I hope our previous response makes it clear that JOOCI is not removing speaker information, form content encoder, but is jointly optimizing both the speaker (other) and content information [1]. This is the reason for its high performance on both speaker and content based tasks as shown in Table 1. JOOCI does not have to pick one information over other. In contentvec and SPIN, the authors choose content over other (speaker) information.
>
> * **Reviewer:**  "Even the MS-HuBERT model used for initializing JOCCI is missing from Table 1."
>     * When the shared and content encoders are frozen, the results for the content based tasks are effectively as MS-HuBERT. For the reader’s reference, the corresponding performance on ASR and PR tasks is provided in Table 5. To ensure clarity, we will revise the table caption in the manuscript to explicitly highlight this equivalence.
>
> * **Reviewer:** "JOCCI relies on a pretrained method RDINO for training the other encoder whereas baseline methods such as HuBERT do not."
>     * **Response:** I hope our previous response [2] clarifies that gains, as shown in Table 1, are not solely due to data volume but also reflect JOOCI's ability to jointly optimize content and other information effectively.
>
> 1. As we mentioned earlier, SPIN can be applied on MS-HuBERT to further improve its content representation since the phonetic content resides in top layers which is not the case of data2vec as explained by the authors of SPIN paper in section 3.2.
> 2. To address this concern, we kindly request the reviewer to refer to Table 1, specifically the last row comparing WavLM+, which is trained on 94,000 hours of data (compared to just 960 + 2500 hours). The comparable performance on SID (a little improvement) and ASV (a little degradation) demonstrates that the gains are not solely due to data volume but also reflect JOOCI's ability to jointly optimize content and other information effectively. We will revise the abstract to clarify this distinction to prevent any potential misunderstanding.
>
> -----
> #### We hope this addresses the reviewer's concerns related to weaknesses. Please let us know if you have any further concerns.

---

### Official Review · Reviewer_beXE · 2024-11-03

**Soundness:** 2
**Presentation:** 3
**Contribution:** 2
**Rating:** 3
**Confidence:** 4

**Summary:**

This submission introduces a framework for distinct representation learning of "content" and "other" properties in speech. The authors report improved performance on certain SUPERB tasks compared to other systems. Additionally, the submission includes comparisons with adapters, ablation studies on encoders, data augmentation, and learned representations.

**Strengths:**

* The research community is highly interested in the topic of speech representation learning.
* The proposed method's evaluation on certain SUPERB tasks yielded better results compared to the cited systems.
* The discussions and comparisons presented are technically sound.

**Weaknesses:**

Major issues:

* The model's effectiveness is unconvincing. The baselines cited are outdated and not state-of-the-art, and the model's performances on the semantic tasks are not better.
* The paper's discussion of different model architectures is shallow, limiting its contribution and making it difficult to draw general conclusions.

Minor:

* Figure 1 could be simplified by removing the hyperparameters.
* The discussion of "Data augmentation" in Line 52 seems out of place, as the initial focus was on model architecture for speech representation learning.

**Questions:**

* Can you offer insights into the relationship between SUPERB downstream task performance and model architecture designs?
* How do your results compare to recent speech representation work that has also been evaluated on SUPERB?

---

> ### Author Response · Authors · 2024-11-14
> **Clarifying JOOCI’s Position, Effectiveness, and Comparisons on the SUPERB Benchmark.**
>
> ### We thank the reviewer for their comments and the opportunity to address their concerns.
>
> * We kindly ask the reviewer to specify what they found unconvincing. The baseline (WavLM) we compare to is state-of-the-art on the SUPERB benchmark leaderboard: https://superbbenchmark.github.io/#/leaderboard. We would appreciate it if the reviewer could point to specific models or methods for additional comparison. For semantic tasks, JOOCI’s performance is very close to that of WavLM, and we mention that WavLM’s use of large batch sizes likely contributes to these minor gains. Therefore these small differences on the sematic tasks could be just noise. As shown in Table 6, with using only the layers with higher content information, a property of JOOCI's content encoder, we can close the gap even further on one semantic task (IC) and beat on the other semantic task (SF). For IC task, the difference is of 0.02 only.
>
> * We also request that the reviewer share specific model architectures they believe would strengthen the discussion and comparisons, allowing us to draw more general conclusions in the paper.
>
> * We will address the reviewer’s comment on Figure 1 in the revised version. Since WavLM (previous SOTA on SUPERB) is trained with data augmentation, we apply data augmentation in JOOCI for fair comparison given the smaller, cleaner pre-training LibriSpeech 960 hours data used.
>
> * We will add a section discussing model architectures and their relation to the SUPERB benchmark in the revised version, as suggested by the reviewer.
>
> * WavLM is one of the most recent models aimed at learning representations across orthogonal tasks such as SID and ASR. Compared to WavLM, our method demonstrates significant improvements on both the tasks. Data2Vec, another method, achieves better results on ASR but is pre-trained specifically to maximize content information. Additionally, on the PR task, JOOCI outperforms Data2Vec, showcasing the effectiveness of the content encoder initialization we used.
>
> We hope this addresses the reviewer’s concerns, and we appreciate any further feedback.

---

### Official Review · Reviewer_YxJB · 2024-11-03

**Soundness:** 2
**Presentation:** 2
**Contribution:** 2
**Rating:** 3
**Confidence:** 4

**Summary:**

The paper proposes a self-supervised speech representation model that combines two encoders, one intended to encode linguistic content and the other intended for "other" content like speaker and emotion information, trained with different losses.  The idea is that, by training a single encoder with a single loss, previous approaches have trouble encoding these two types of information equally well.  The various elements of the model are largely borrowed from previous work, but combined in a new way.  The model is compared in terms of performance on 8 common tasks (from the SUPERB benchmark) to other commonly used models (HuBERT, WavLM), finding improved performance on 4 of the tasks.  The paper also includes some ablation studies and analyses of several model components.

**Strengths:**

+ Addresses an important need to account for both linguistic and non-linguistic content in speech representation learning.

+ Obtains impressive results on several tasks, including speech recognition and speaker identification.

**Weaknesses:**

- Presentation of many details is unclear.  For example, the definition of "content" and "other" is never clearly stated.  Also, the model description is very brief, leaving many details to cited papers or the imagination (for example, is prosody ever/always/sometimes considered "content"?).  Either the writing should be much more precise or the paper should include equations specifying all of the model components.  See some other specific questions below.

- The key claimed contribution is that the model encodes both linguistic and non-linguistic information and that these are disentangled into the two encoders' representations.  However, the results don't quite show this, since the results on tasks are mixed and the analyses don't really demonstrate disentanglement (again see questions below).  Overall, I don't see the community starting to use this model as a replacement for other currently popular models.

- Some of the experiments do not, as far as I can tell, show the claimed findings (see details in "Questions" below).

- The writing is in general hard to follow at times, in part due to many grammatical errors.

**Questions:**

- The paper states that WavLM the "previous SOTA method".  By what measure is WavLM SOTA?  On what task(s)?

- I don't follow the sentences "As a result, the model cannot fully leverage all layers ... within a single embedding." nor the following sentence "The strategy of dividing the layers..."  Can you clarify what is meant there?

- The description of the split and append layer is a bit hard to follow.

- In Eq. 1, the index d is never used in the summand.  Also, should "MPL" be "L_MPL"?

- In Eq. 3, what exactly are Student^PN and Teacher^RDINO?

- In Table 1, where are the results for FBANK and other competitor methods obtained from?  Citations should be provided.  I also suggest including MS-HuBERT since JOOCI is based on it, and ideally also data2vec which has good results on many SUPERB tasks (but please let me know if you think these would not be relevant for some reason).

- I don't quite follow the sentence "We augment the data very lightly, so not to interfere with the content encoder a lot and divide its capacity."

- The description of the main results in Section 3.2 seems a bit misleading.  The paper states that the "results clearly indicate that JOOCI outperforms the current state-of-the-art (SOTA) models on the majority of tasks, except few ...".  However, in Table 1 JOOCI appears to outperform other models on exactly half the tasks, and it is never explained in what sense those models are SOTA (though they are clearly commonly used models).

- I do not understand the purpose of the comparison in Table 2, since JOOCI is not an alternative to adapters.  Also, "Houlsby" and "CHAPTER" need to be defined.

- How does Figure 2 show the effect of data augmentation?  Is there a pair of curves that differs only in the use of data augmentation?

- For Figure 2, more information is needed about the y-axis.  How is CCA similarity defined?  How are the word labels encoded and how many words are there?  There has been prior work using CCA similarity for layer-wise analyses, e.g. Pasad et al., "Comparative layer-wise analysis of self-supervised speech models," ICASSP 2023.  Figure 2 seems similar to some of this prior work, and so it would also be helpful to state how your CCA-based analysis is the same or different, and whether your HuBERT results are similar to Pasad et al.'s.

- The ablation study in Sec. 4.1 is a bit confusing to me.  It claims to separately show the effect of DGRL and data augmentation, but as far as I can tell these two variables are changed simultaneously in the experiments.

- In Table 3, why are the "-" results not included?  If those could be included, they could help to show to what extent JOOCI-C and JOOCI-O specialize for linguistic vs. non-linguistic information.

- In Section 4.2, I have trouble following the first paragraph.  What kind of information is considered "higher-level" in the "other" branch, and what is the "same trend" that is referred to here?

- Section 4.3 claims to "prove that JOOCI is able to disentangle content and other information", but I don't follow how the results show this.  (Also, the word "prove" is too strong here, as in most descriptions of empirical findings.)

- In Table 5, what is the difference between the experiments in the last two lines (both labeled "JOOCI (6-11)"?

---

> ### Author Response · Authors · 2024-11-28
> **Clarification on the presentation and writing.**
>
> ### We sincerely thank the reviewer for their detailed questions. They were very helpful.
>
> * Firstly, we would like to clarify that JOOCI does not aim to remove other information from the content or employ a disentanglement module. Instead, JOOCI jointly optimizes content and other information by using separate paths/encoders, focusing on maximizing representation quality rather than enforcing a strict separation, as seen in WavLM.
>     * We have discussed this in detail with reviewer "rAqx" for your kind reference. If there are still any concerns or points of dissatisfaction, please let us know, and we would be happy to address them further.
>
> * We have worked to improve the writing and presentation in the revised version, addressing a lot of questions raised by the reviewers to provide better clarity and structure overall. We will aim to still make it better if the reviewer still has questions and suggestions.
>
> -------
>
> #### Once again, thank you for your thorough feedback, which has been invaluable in refining the manuscript and enhancing its clarity and presentation for future readers.

---

> > ### Comment · Reviewer_YxJB · 2024-11-30
> >
> > Thank you for the responses and revisions.  I think the process has helped to clarify the contribution of the work.  Besides the point about disentanglement, however, I cannot find responses to all of my questions/comments in the revised version or in the responses to reviewers.  However, I may be missing something.  If you could point out where in the revised version each of my questions is addressed, or respond here to the ones that aren't, I would be happy to consider revising my review.  As of now I remain happy with my initial rating.

---

> ### Author Response · Authors · 2024-11-30
> **Thank you again for helping to improve the paper.**
>
> ### Weaknesses:
>
> * Presentation of many det....
>     * We will add a subsection in section 2 in the final version, explaining clearly what is meant by other and content information present in the input speech accompanied by a diagram or table. This section will be before the component subsection (section 2.1) so that the readers have an understanding of what different encoders are doing.
>
> * The key claimed contribution is that ...
>     * We have not claimed any active disentanglement, but it was an observation made by us, given the empirical results of Table 5 and to some extent Table 4.
>         * When using JOOCI-O-DGRL, the performance on the PR (content task) is close to 100%. And when using JOOCI-C, the performance of the SID task is poor. JOOCI-C maximizes the content information and JOOCI-O-DGRL is trained to maximize other (speaker information using the RDINO as teacher).
>     * SInce the other in JOOCI is not just to encode speaker information. The DGRL module is used as a regularizer to avoid overfitting on the speaker based tasks (and not to be seen as a disentanglement module to improve the performance on the speaker tasks). The DGRL module resulted in improved performance on the emotion recognition task and slight reduction on the speaker based tasks.
>
> * The writing is in general hard to...
>     * The reviewer is right and we are thankful for all the comments.  We have tried to revise the writing part and hope to revise it again after collecting all the feedback during the rebuttal phase.
>
> ---------------------------
>
> ### Questions:
>
> * The paper states tha...
>     * This notion, of WavLM as a SOTA when evaluated on a range of tasks, is well accepted in the speech community.  The reviewer can confirm the same from  the official SUPERB benchmark website. For some reason the link is down (https://superbbenchmark.org/leaderboard). Therefore we request the reviewer to please use the backup link: https://superbbenchmark.github.io/#/leaderboard. The WavLM series of models are at the top.
>     * SUPERB is a leaderboard to benchmark the performance of a shared model across a wide range of speech processing tasks with minimal architecture changes and labelled data. The official paper is at: https://arxiv.org/pdf/2105.01051
>
> * I don't follow the senten...
>     * Please re-read from line no 82 in the revised version. Though we recommend reading the introduction section again starting line no 41.
>
> * The description o...
>     * We have tried to show the working of split and append layer in Figure 1. We will add a dedicated paragraph with the heading split and append layer in the components section (section 2.1) discussing it in detail.
>
> * In Eq. 1, the index d is ...
>     * We are unable to understand the question and request further clarification.
>
> * In Eq. 3, what exactly ....
>     * PN and RDINO are the student and teacher modules used. We have tried to clarify it in line 190. We will clarify further if the reviewer recommends.
>
> * In Table 1, where are...
>     * We will add the citations as suggested in the final version.
>     * MS-HuBERT and data2vec are shown to maximize content information. The same is observed with their superb performance on the ASR and PR task. Both focus solely on maximizing content information, while JOOCI aims to jointly optimize both content and other information, aligning more closely with WavLM's objectives. WavLM used data augmentation to boost the performance on the other tasks such as speaker and emotion.
>     * When the shared and content encoders are frozen, the results for the content based tasks are effectively as MS-HuBERT. For the reader’s reference, the corresponding performance on ASR and PR tasks is provided in Table 5. To ensure clarity, we will revise the table caption in the manuscript to explicitly highlight this equivalence.
>
> * I don't quite follow the senten....
>     * We have tried to clarify this in the revised version and sincerely request the reviewer to please read section starting line no 244. When fine-tuning the shared and content encoder with Strong data augmentation, the content encoder would behave similar to WavLM i.e., fewer layers will be used to encode content information. Since the earlier layers would try to remove the noise from the audio.
>
> * The description of the....
>     * We would like to clarify that the difference in performance is miniscule compared to WavLM and the reason could be very large batch sizes used by WavLM during the evaluation. This is mentioned in the WavLM paper TABLE IX (https://arxiv.org/pdf/2110.13900) and is not recommended to change as was shared by the authors of SUPERB in one of their issues: https://github.com/s3prl/s3prl/issues/360#issuecomment-1155008924. Therefore these small differences on the sematic tasks could be just noise
>     *  Furthermore, we request the reviewer to please check Table 4. The gains can be reduced further and even be surpassed by JOOCI when using layers with higher content information.

---

> ### Author Response · Authors · 2024-11-30
> **Contd.**
>
> * I do not understand....
>     * We will move this discussion to the appendix. We showed the results so that the readers could have a bigger point of view of JOOCI vs HuBERT with adapters. In no capacity we suggest it is a replacement of adapters. On the other hand, adapters can be applied to JOOCI also. We have revised the writing and request the reviewer to pleas re-read from line no 288.
>
> * How does Figure 2 show..
>     * MS-HuBERT was trained WITHOUT data augmentation and is used as an initialization for JOOCI. Later, the content encoder of JOOCI is finetuned WITH data augmentation. As shown in Figure 2, the later layers have shown a increase in CCA score.  I hope it clarifies.
>
> * For Figure 2, more information is ne.....
>     * We apologize to the reviewer. Somehow, we missed to cite the paper. In the revised version we have mentioned it properly. To clarify, the analysis is exactly the same.
>
> * The ablation study in Sec. 4.1 is ....
>     * We first study the effect of DGRL module on the representations learned by the other encoder, while keeping the shared and content encore frozen without using data augmentation.
>         * With and without DGRL results are shown for this config. DGRL module aligns the other encoder better with the claim of JOOCI i.e., to jointly optimize OTHER and content information. Other not just means speaker.
>     * We have tried to improve the writing and request the reviewer to please re-read the section 4.2. Please tell us what is still missing and we will add the information in the revised version.
>
> * In Table 3, why are th...
>     * They are skipped because (i) won’t provide any additional information (ii) limited compute available. In the revised version, Table 5 last row shows the extent to which JOOCI-O-DGRL encodes both the content (PR) and other (SID) information. Close to 100% for the PR task, this means no meaning full content information is encoded in the other encoder.
>
> * In Section 4.2, I have trouble followin....
>     * We request the reviewer to please re-read this section in the revised version for better clarity. It is section 4.1 line no 353.  The results are updated. We made a minor mistake in earlier setup.
>         * Please ignore the numbers in asterik them for now. They show the performance when using multi-head attention.
>     * Same trend means: earlier layers having higher weight.
>
> * Section 4.3 claims to "prove that....
>     * We will tone down the claims and make it clear that it is an observed behavior of JOOCI, and is not pre-trained for.
>         * Using later layers of content encoder, which has high CCA score, resulted in high ASV score, compared to using all the layers. This shows that the interference of speaker is very minimum for these layers for the VC-a2a task . This led us to claim that the content encoder (later layers) are able to disentangle content information from other information (speaker).
>         * Similarly the performance of the tasks which require content information also improved, Table 4, when using only the high CCA score layers.
>
> * In Table 5, what is the d....
>     * In the revised version, it is Table 6. We have added the context.
>
> ---------------------
>
> #### Thank you again. Looking forward to your valuable feedback.

---

> > ### Author Response · Authors · 2024-12-02
> >
> > A gentle reminder.

---

> > > ### Comment · Reviewer_YxJB · 2024-12-03
> > >
> > > Thank you for the additional information!  There are some points that are yet to be revised to improve the clarity of the paper, and I look forward to seeing a future version of the paper.  For now I am still happy with my score.  I don't have time to respond to all of the points individually, but here are replies regarding a few of them:
> > >
> > > * I do not understand....
> > >   - This notion, of WavLM as a SOTA when evaluated on a range of tasks, is well accepted in the speech community...
> > >
> > > I agree that WavLM is a solid model family that does well on many tasks, and it is reasonable to compare to it.  However, I am not sure what you mean by "well accepted in the speech community".  "SOTA" implies that it has the best existing result by some metric.  When I look at the SUPERB leaderboard, for example for the "challenge hidden dev" setting, I see that the best model differs across tasks.  For many tasks it is WavLM Large, but for some tasks it is WavLM Base+, WavLM Base, or wav2vec 2.0 Large.  If by "SOTA" you mean the best according to some overall metric like the SUPERB "rank" or "score", then you should say so and then compare your model using that metric.  If instead you want to compare performance over several individual tasks (as the paper is currently doing), then you should find the best result for each task in the literature and compare to it.
> > >
> > > I believe the SUPERB leaderboard does not include all of the latest models, but rather only a set of models that were evaluated at a particular time.  Finally, SUPERB has a particular way of evaluating models using a constrained prediction head, and it may be possible to obtain better results using a more complex prediction head, fine-tuning, etc. (see, for example, https://arxiv.org/abs/2306.00452).
> > >
> > > Overall, I think it's fine to compare to WavLM, but "SOTA" is not the correct description for it.
> > >
> > >
> > > * I don't follow the senten...
> > >   - Please re-read from line no 82
> > >
> > > Thanks for revising.  I'm afraid I still don't know exactly what is the problem that is being pointed out with prior approaches:  if they encode content information in some layers and "other" information in other layers, this means that both kinds of information are encoded *somewhere* and that the two kinds of information are (at least to some extent) disentangled.
> > >
> > >
> > > * In Eq. 1, the index d is ...
> > >   - We are unable to understand the question and request further clarification.
> > >
> > > Eq. 1 is L_CL = \sum_d (MPL).  My point is that (MPL) does not appear to depend on d, and in addition "MPL" is not defined but seems to be later replaced by L_MPL.

---

### Official Review · Reviewer_9Wd2 · 2024-11-10

**Soundness:** 2
**Presentation:** 1
**Contribution:** 1
**Rating:** 3
**Confidence:** 5

**Summary:**

This paper proposes a method for speech representation learning, particularly, for disentangle content information from non-content information. The paper reported strong experimental results on SUPERB benchmark.

**Strengths:**

The proposed method is sound. The experimental results on a subset of SUPERB benchmark are strong.

**Weaknesses:**

- The novelty is limited. The proposed method is very close to a number of existing works, e.g.:
  - Chan et al., Content-Context Factorized Representations for Automated Speech Recognition, InterSpeech 2022.
  - Zhao et al., CCSRD: Content-Centric Speech Representation Disentanglement Learning for End-to-End Speech Translation, EMNLP 2023.
- The main claim is flawed. The paper claims SOTA on SUPERB. However, it only reports experimental results on a subset of the tasks from SUPERB (7 out of 10).
- The writing needs improvements:
  - Importantly, the name "other encoder" is a poor choice, which causes a lot of confusion for reading. Some simple choices such as "non-content encoder" would do a much better job.
   - Secondly, many small claims are questionable throughout the paper. A few examples:
     - Abstract: content and non-content information are orthogonal -- in the words from the paper, “how it is expressed” depends on “what is being said”
     - Sec 2.2: "Since JOOCI uses separate learnable parameters, the losses are summed directly without requiring
additional hyperparameter tuning." -- The previous paragraph said the opposite: " The GRL scale the gradients during backpropagation by a factor of 1/10, preventing interference with the other loss."
  - Lacks details of the model. While references to prior works is great, for completeness of the paper, you should describe the details of you model clearly, so that the readers understand your approach without having to jumping to many other papers.

**Questions:**

See Weaknesses section.

---

> ### Author Response · Authors · 2024-11-13
> **Addressing Misunderstandings Regarding JOOCI’s Methodology and Novelty.  JOOCI is not doing any disentanglement.**
>
> ### We sincerely thank the reviewer for their comments and for the opportunity to clarify certain aspects of the JOOCI framework's design and novelty.
>
> * We will clarify the main claim and clearly mention specific tasks instead of the using the SUPERB benchmark term.
>
> * Regarding the reviewer's remarks on JOOCI's novelty and its alleged similarity to previous work, we would appreciate further clarification. The reviewer has highlighted two specific papers, which both employ a disentanglement module and use supervised data during training in some way. However, JOOCI is fundamentally different: it does NOT include any disentanglement module, nor is it trained with supervised data during pre-training. This distinction contrasts with the reviewer’s assertion about JOOCI’s similarity to the referenced works.
>
> * Additionally, we respectfully note that the reviewer's summary of JOOCI appears to contain a misunderstanding. Specifically, JOOCI is not designed as "speech representation learning for disentangling content information from non-content information," as the summary suggests. In fact, the concept of disentanglement is only mentioned in Section 4.3 of the paper (in the second-to-last section before the conclusion) as part of an ablation study on understanding learned representations. This brief remark reflects an observed behavior in the model’s learned representations rather than a deliberate design goal or imposed structure during pre-training.
>
> * We will add the details of the model as suggested. Lastly, as suggested we will improve the writing of the paper. We have updated the rebuttal, which does address this issue to a major extent.
>
> We hope this clarification addresses the reviewer's concerns and would be grateful for any additional feedback they might provide on this matter.

---

> > ### Comment · Reviewer_9Wd2 · 2024-11-13
> >
> > Thank authors for the response.
> >
> > Unfortunately, I'm confused by the author response, stating that my summary misunderstood the intention of the paper. "JOOCI is not designed as "speech representation learning for disentangling content information from non-content information," " -- isn't that the motivation stated in the abstract of the paper?

---

> > > ### Author Response · Authors · 2024-11-14
> > > **Response**
> > >
> > > ### We thank the reviewer for their follow-up and appreciate the opportunity to clarify further.
> > >
> > > * The motivation described in the abstract is focused on optimizing the joint representation of content and other information by leveraging two separate paths/embeddings within a self-supervised learning framework. This approach is not intended for explicit disentanglement but rather for learning orthogonal representations without imposing a separation of information. This is different to current methods which encode other information in the earlier layers of the encoder, resulting in the model dividing its total capacity, limiting their ability to build complex hierarchical features effectively for other information (characteristic for deep learning) . Our results show that JOOCI is able to achieve competitive performance on both PR and SID tasks. This is different from previous methods, such as data2vec which maximize PR performance (data2vec) while lagging behind SID. Or WavLM which finds a local minima for PR and SID (dividing its total capacity). JOOCI achieves better performance jointly on PR and SID compared to earlier methods.
> > >
> > > * If there are specific lines in the paper that may have implied a disentanglement motivation, we would be grateful if the reviewer could point them out. This feedback would help us address any potential ambiguities and refine our presentation accordingly for future.

---

### Author Response · Authors · 2024-12-02
**Clarification on the contribution of JOOCI to the speech research community.**

### The idea of developing a single end-to-end model to __JOINTLY__ learn comprehensive speech representations that can effectively handle various downstream tasks jointly is both essential and a important research direction in the speech community.
#### There have been models in the past that perform well on certain tasks but poorly on others. Specifically, models that excel in content tasks, such as phoneme recognition (PR), tend to perform poorly on speaker tasks, such as speaker identification (SID).
-----
In the research community, WavLM stands out as a widely popular model designed to learn comprehensive speech representations by leveraging (i) masked predictive learning (MPL) to capture content information and (ii) data augmentation to encode other aspects of speech. WavLM demonstrated superior performance __JOINTLY__ on content and speaker tasks compared to HuBERT.
* The authors observed that WavLM achieved this by dividing its total modelling capacity/layers, with later layers learning content and earlier layers learning other information.
* A major flaw is that this ultimately prevents any model from fully leveraging all layers to build the complex, hierarchical representations characteristic of deep learning.
----
Our proposed JOOCI framework make use of all the layers (depth) available to build hierarchical representations. This design choice resulted in remarkable performance improvements __JOINTLY__ on content and speaker tasks compared to WavLM as shown in Table 1.
* JOOCI's impressive __JOINT__ performance on these orthogonal tasks has been prominently acknowledged by
__unanimously__ by all the  reviewers.
* Our discussion with reviewer __rAqx__ also shows that the gains are not solely due to data volume but also reflect JOOCI's ability to jointly optimize content and other information effectively.  JOOCI achieves comparable performance with WavLM+, which is trained on 94,000 hours of data.

Finally, we believe that JOOCI takes the speech community one step closer in developing a single end-to-end model capable of __JOINTLY__ learning comprehensive speech representations on various downstream tasks.

-------

---

### Meta-Review · Area_Chair_5Lbn · 2024-12-18

**Metareview:**

The paper proposes an approach to disentangle linguistic information from non-linguistic ones by using an additional speaker model for supervision.

I recommend a rejection because of  the unsupported claims in the paper.

All reviewers raised concerns about unsupported claims. Whether the representations are properly disentangled is not well supported, not to mention disentanglement by itself is not a very well defined concept. Whether it is theoretically possible to disentangle linguistic and non-linguistic information is another inherent claim that is not supported.

The biggest argument is around the term "state of the art." Most reviewers are not happy with the claims involving the state of the art. I personally don't mind using the term to refer to the current "state." However, the paper is using the term to refer to the "best." It becomes tricky to claim the best because there isn't a single metric in the evaluation. (It's also generally not useful to talk about the best in this context anyways.) It's much more convincing to claim victory over a strong baseline, rather than being the best. Besides, since the paper uses a speaker model, it might not even be fair to claim victory over those that don't.

Overall, a lot of care is needed in the experiments and in the writing.

**Additional Comments On Reviewer Discussion:**

The discussion had been on point and healthy. However, the authors failed to address all the questions during the rebuttal. The main problem as summarized in the metareview was around the attribution of improvements.

---

### Decision · Program_Chairs · 2025-01-22

Reject